Subject Area:
cellular biology

Keywords:
thylakoid biogenesis, thylakoid membrane, chloroplast, chloroplast vesicle transport, resurrection plants

Author for correspondence:
Jürgen Soll
e-mail: soll@lmu.de

† This paper is dedicated to Prof. Dr Gernot Schultz on the occasion of his 90th birthday.

# A brief history of thylakoid biogenesis†

Annabel Mechela[1], Serena Schwenkert[1,2] and Jürgen Soll[1,2]

[1]Department Biologie I, Botanik, Ludwig-Maximilians-Universität, Großhaderner Strasse 2-4, 82152 Planegg-Martinsried, Germany
[2]Munich Center for Integrated Protein Science CiPSM, Ludwig-Maximilians-Universität München, Feodor-Lynen-Strasse 25, 81377 Munich, Germany

SS, 0000-0003-4301-5176

The thylakoid membrane network inside chloroplasts harbours the protein complexes that are necessary for the light-dependent reactions of photosynthesis. Cellular processes for building and altering this membrane network are therefore essential for life on Earth. Nevertheless, detailed molecular processes concerning the origin and synthesis of the thylakoids remain elusive. Thylakoid biogenesis is strongly coupled to the processes of chloroplast differentiation. Chloroplasts develop from special progenitors called proplastids. As many of the needed building blocks such as lipids and pigments derive from the inner envelope, the question arises how these components are recruited to their target membrane. This review travels back in time to the beginnings of thylakoid membrane research to summarize findings, facts and fictions on thylakoid biogenesis and structure up to the present state, including new insights and future developments in this field.

## 1. Introduction

Life on Earth largely depends on atmospheric oxygen that is produced by photosynthetic cyanobacteria and plants. Oxygenic photosynthesis is estimated to have evolved 3–3.5 billion years ago, thereby enriching the Earth's atmosphere with oxygen [1,2]. Alongside the emergence of photosynthesis and aerobic respiration, mitochondria as the primary site for respiration and chloroplasts as the place of photosynthesis were established as new cell organelles. Both originated independently from a primary endosymbiotic event during which a prokaryote was engulfed into the cytoplasm of a host cell [3]. In the case of chloroplasts, a eukaryotic host that already contained mitochondria took up an ancestor of today's living cyanobacteria which massively transferred parts of its genome to the host nucleus to eventually become a cell organelle still containing its own plastid DNA [2,4–7].

Owing to their endosymbiotic origin, chloroplasts are surrounded by two envelope membranes, both of which are of prokaryotic origin. In addition, chloroplasts possess an internal membrane network lying in the aqueous stroma called the thylakoids which are the place of the light-dependent reactions of oxygenic photosynthesis. Owing to the unique composition and structure of the thylakoid membrane, solar energy is efficiently converted into chemical energy with the help of major protein complexes. Chlorophyll pigments in the multi-subunit protein complexes of photosystem II (PSII) and photosystem I (PSI) are excited by light and initiate electron flow between the complexes. This process generates chemical energy in form of ATP and reducing equivalents such as NADPH that later are used in the light-regulated reactions of the Calvin–Benson cycle located in the stroma. There, $CO_2$ is fixed by the abundant enzyme ribulose-1,5-bisphosphate carboxylase/oxygenase (RuBisCO) to supply the cell with carbohydrates [8,9].

royalsocietypublishing.org/journal/rsob    Open Biol. 9: 180237

## 2. Chloroplast differentiation—in the beginning is the proplastid

In angiosperms, plastids develop during a light-dependent differentiation process from a simple undeveloped progenitor called a proplastid. Proplastids are non-photosynthetic small round-shaped organelles that are present in meristematic tissues of the shoot as well as the root apex [10]. They contain only few internal membranes mostly in the form of vesicles or small saccular structures that sometimes are in contact with the inner envelope [11–13]. The maturation process starts with the formation of long lamellae inside the proplastid. Later, these lamellae are transformed into disc-shaped structures that assemble into grana stacks. Eventually, the complex and intertwined thylakoid membrane network arises in mature chloroplasts (figure 1a–f) [8].

In the middle of the twentieth century, Strugger contributed to the idea that chloroplasts would continuously emerge from proplastids. Proplastids were thought to be amoeboid organelles with a granular stroma capable of assimilating starch out of sugar. The primary granum was described as a chromo-nucleo-proteoid complex consisting of vesicles and lamellae. Duplication of the primary granum and subsequent enlargement of the stroma in meristemic tissues would finally result in the formation of non-amoeboid juvenile chloroplasts characterized by a lens-shaped flattened appearance and the existence of parallel layered secondary grana, but smaller than mature chloroplasts [15,17,18]. Later, it was clarified by Gunning & Jagoe [19] that the described primary granum was the same as the prolamellar body that can be observed in etioplasts, a plastid form occurring in the absence of light in angiosperms. A more detailed view on etioplasts and their internal structure is given in the course of this review.

According to von Wettstein [18], several processes were thought to be necessary for the formation of the chloroplast structure. Among these were protrusions and folds, thickening and splitting, fusion and division of the lamellae. Strugger's theory of the continuity of a primary granum was later replaced by the hypothesis of a homogeneous proplastid surrounded by a double membrane [11,20]. Furthermore, it was observed that all lamellae found in chloroplasts were built of tubules that for their part derived from the inner envelope as invaginations (figure 1c) [11,16,21].

At the same time, Menke [22] speculated whether alternative secondary pathways for thylakoid formation such as regeneration inside the stroma, disaggregation and rebuilding or invaginations of the thylakoids themselves could exist. Additionally, Wehrmeyer & Röbbelen [23] considered processes of overlapping growth of lamellar structures to contribute to thylakoid formation. At the end of the 1960s, it was debated whether the thylakoid membrane system and the inner envelope could be discontinuous or rather were a connected system. It had been considered that a continuous membrane system would only be present in early stages of differentiation and absent during later stages [24,25].

It was also suggested that the differentiation process in angiosperms is light-induced rather than dependent on the cell cycle. In the 1950s, Fasse-Franzisket [26] was one of the first to investigate the influence of different light intensities and wavelengths during greening. In the absence of light, proplastids are not able to form chloroplasts but instead turn into etioplasts. These organelles contain a spherical and paracrystalline structure of about $1-2\,\mu m$ in diameter called the prolamellar body (PLB), which consists of plastid lipids, the chlorophyll $a$ precursor protochlorophyllide (Pchlide) and the NADPH- and light-dependent enzyme protochlorophyllide oxidoreductase (LPOR) [27–31]. The PLB was first observed in 1953 by Leyon [32] and described as a 'dense core'. Subsequent terms were introduced such as 'primary granum' [33], 'plastid centre' [18], 'vesicular centre' [34] and 'Heitz-Leyon crystal' [35]. Electron microscopy revealed the PLB as a compound structure made up of membranous lattices resulting in a mosaic appearance resembling a crystalline form [19]. The special paracrystalline symmetry observed in PLBs is achieved by complex formation of its major components. LPOR accumulates with Pchlide and NADPH in a regular manner forming the so-called protochlorophyllide holochrome [36] to ensure protection against proteolysis during darkness [37]. The function of PLBs is not fully understood to date, but it could act as a storage place for lipids and proteins that are needed for the synthesis of the photosynthetic apparatus as light becomes available [31,38]. From the crystal-like centre of the PLB, perforated tubular lamellae reach out into the stroma or even connect to the inner envelope. These structures are named prothylakoids (PTs) as they strongly resemble unstacked stroma lamellae [31].

Light induces the transition from the etioplast stage to a mature chloroplast via dispersion of the PLB [13,27]. Upon illumination, the PLB disintegrates rapidly and simultaneously with the photochemical reduction of protochlorophyllide to chlorophyllide. It was observed that the PLB first enlarges in size before it disperses into small spherical vesicles [31]. Moreover, it was suggested that these vesicles would then arrange in primary layers to eventually fuse into discs to form grana [39,40]. Interestingly, a new PLB will form upon re-darkening of chloroplasts [41].

In contrast to this view, tubular structures of bean PLBs were shown to transform directly into flat slats without dispersing into vesicles. These slats then continuously formed first stacked grana structures through overlapping of neighbouring membranes. Thylakoid formation furthermore coincided with the observation of the appearance of the first chlorophyll–protein complexes. This indicated protein complex arrangement and membrane formation as a crucial interplay for chloroplast biogenesis [42].

Photomorphogenesis, in general, is a highly coordinated process that requires numerous cellular changes. Light perception via photoreceptor proteins like phytochromes and cryptochromes initiates chloroplast biogenesis via alteration of gene expression, import of nuclear-encoded proteins, increase of chlorophyll content and finally the establishment of a thylakoid network. Already, in the 1980s, it was experimentally proved that photomorphogenesis of etioplasts towards mature chloroplasts was triggered by red as well as blue light [43]. Phytochromes respond to red and far-red light, while cryptochromes perceive blue and UV light [44].

In angiosperms, photomorphogenesis of proplastids towards mature chloroplasts takes place at the vegetative shoot apex. This region can therefore be considered as the initiation site for thylakoid biogenesis. There, a layered structure called the shoot apical meristem (SAM) and flanking leaf primordia can be found. The SAM can be grouped into a central zone (CZ) at the tip, a peripheral zone (PZ) that surrounds the CZ and a rib zone (RZ) that is found beneath

royalsocietypublishing.org/journal/rsob    Open Biol. **9**: 180237

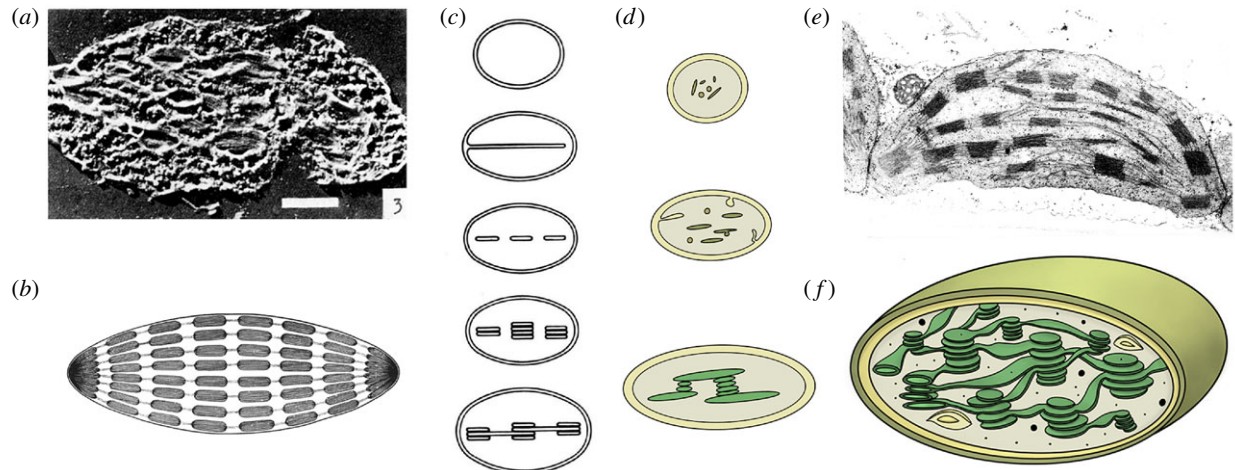

**Figure 1.** Chloroplast structure and development in the past and today. Already at the beginning of the 1950s, the first electron microscopic pictures of chloroplasts of land plants were taken. (*a*) The first cross-section through a tulip chloroplast [14]. (*b*) The inside of chloroplasts was considered to consist of stroma and connected grana thylakoids that were believed to be arranged like rolls of coins [15]. (*c*) Differentiation towards mature chloroplasts was suggested to happen from a progenitor via invaginations of the inner envelope. Invaginations would then split into disc-shaped vesicles that stacked together to be eventually interconnected [16]. (*d*) The modern view on chloroplast differentiation is very similar to that of the past. The progenitors are proplastids that contain only few internal membranes and vesicles that finally assemble the thylakoid membrane network in the presence of light. (*e*) Modern electron micrographs provide insights into the actual arrangement of the grana stacks. The complex internal organization of a chloroplast is depicted in (*f*) with thylakoids forming a highly interconnected fretwork.

the CZ. The CZ contains stem cells that later will be forming all aerial parts of the plant. The PZ is the source of cells from which leaves develop while the RZ provides cells for the internal tissues of the stem and leaves. Beside this classification, the SAM can also be subdivided into three distinct layers called L1, L2 and L3, each of which generates different parts of the leaf. L1 and L2 constitute the epidermis and outer mesophyll, whereas L3 contributes cells for the inner mesophyll and vasculature. For a long time, it was thought that the SAM would only harbour proplastids, while primordial leaves would already contain mature chloroplasts. A developmental gradient was thus predicted to exist between the two regions of the shoot apex. In contrast to these beliefs, it could be shown that the SAM was not at all uniform regarding chloroplast differentiation. L1 and L3 layers contained plastids that already possessed small thylakoid networks and chlorophyll-binding proteins, whereas L2 layer plastids totally lacked thylakoids [45]. Not only in the SAM but also in growing leaves, chloroplasts show clear developmental gradients. These gradients can be observed not only between leaves of different age but also within a given single leaf. Leaves at the tip of the shoot are the first to complete the differentiation process, while leaves at the lamina base are the youngest. However, an age gradient exists not only from top to base but also from the leaf margin to the midrib [46]. Taking this into account, chloroplast differentiation in dicotyledonous plants is not as homogeneous as in monocots that exhibit a gradient along the leaf blade. Grasses, for example, show a gradual development of plastids from the leaf base, the location of the meristem, to the leaf tip where mature chloroplasts reside. The zoning of monocotyledonous leaves had already been observed in the 1950s. Even then, the different visible pigmentation of the leaves was associated with the idea of a developmental gradient from proplastids to chloroplasts. Young stages of development were found at the leaf base while differentiated chloroplasts increased towards the leaf tip. It could also be shown that the number of plastids increased with ongoing differentiation [26]. In dicots, this process seems to be more complex as it is not only dependent on the leaf part but additionally on the organ, tissue and developmental stage of the plant [47].

## 3. From rolls of coins to reality— unravelling thylakoid morphology

Investigation of thylakoid biogenesis and structure is closely related to the light-dependent differentiation process from proplastids to mature chloroplasts. The thylakoid membrane is a unique feature of organisms that perform oxygenic photosynthesis. In contrast to prokaryotes performing anoxygenic photosynthesis, where the internal membranes are continuous with the plasma membrane [48,49], thylakoids in mature chloroplasts as well as in cyanobacteria no longer seem to be connected to the inner envelope or the plasma membrane, respectively. In general, the thylakoid structure in mature chloroplasts is more complex than in cyanobacteria and many algae, which mainly contain single layers of long lamellae (figure 2*a*,*b*). In chloroplasts from land plants, thylakoids appear as an intertwined network of stroma lamellae and densely packed interconnected grana stacks providing a huge surface for metabolic processes (figure 2*c*) [8].

Already very early in the history of chloroplast research, the focus was laid on investigating the detailed structure of this complex network. The first observations of a granular distribution of chlorophyll inside chloroplasts were made in the middle of the nineteenth century [53,54]. This led to a first structural differentiation of the chloroplast in a colourless stroma [55] and green pigmented grana [56,57], which contrasted with the opposed belief in a homogeneous composition of the organelle [58]. In the 1930s, several publications confirmed the granularity hypothesis [59,60] and paved the way for a deeper understanding of chloroplast structure. Furthermore, it could be shown that the described granules were not globular-shaped but rather depicted a slice-like form [60]. The arrangement of these grana slices was debated to be helical [61], irregular [60] or like rolls of coins, forming

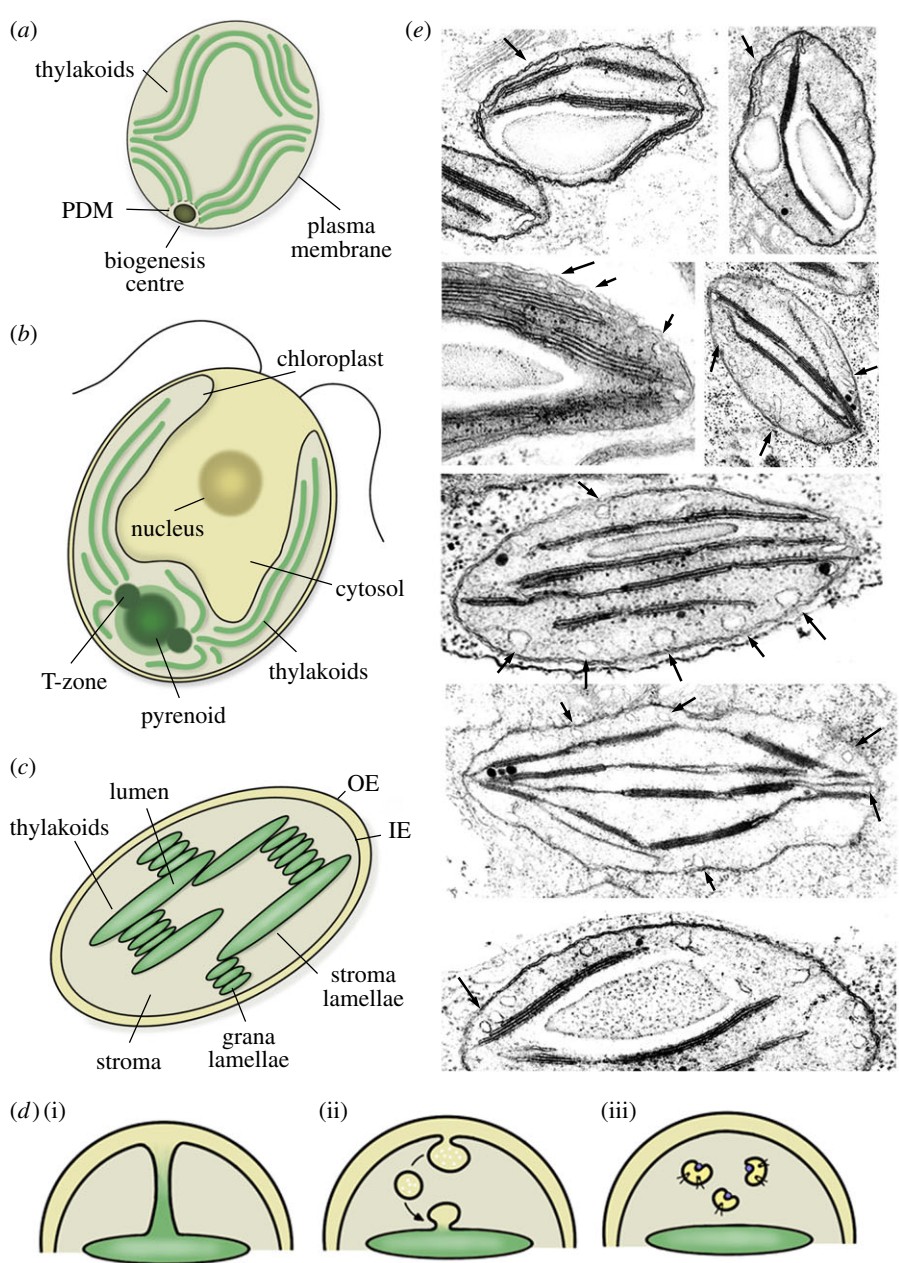

**Figure 2.** Thylakoid structure and biogenesis in cyanobacteria, green algae and land plants. Thylakoids of cyanobacteria and green algae are less complex than those of land plants. (*a*) Cyanobacterial thylakoids consist of single lamellae. Specialized membrane regions are believed to function as thylakoid biogenesis centres. These PDMs appear at convergence zones of the thylakoid and the plasma membrane [50]. (*b*) Green algae contain only one single chloroplast with concentric thylakoids. The pyrenoid, a plastid microcompartment, supports with the fixation of $CO_2$. Around the pyrenoid, the translation zone is also believed to function in thylakoid biogenesis [50,51]. (*c*) Chloroplasts of land plants differ from those of cyanobacteria and green algae as they build an intertwined network of stroma lamellae and grana lamellae. (*d*) As it is known that many important building blocks for the thylakoids derive from the inner envelope, three ideas exist on how this transfer could happen. Components could either bridge the stroma via invaginations from the inner envelope as shown in (i). Alternatively, they could travel as cargo of plastid vesicles (ii) or be shuttled by soluble transfer proteins (iii, proteins and lipids attached to a cargo protein are shown) [52]. (*e*) Arrows pinpoint vesicles and invaginations as seen in young pea chloroplasts. The upper part of the figure shows tubular invaginations extending from the inner membrane to the thylakoids. The lower part shows plastic vesicles, which occur freely in the stroma and are no longer connected to the inner membrane. In the bottom panel, one can see stromal vesicles docking to the thylakoid membrane.

an internal lamellar system [15,62]. Early hand drawings of chloroplast cross-sections illustrate grana stacks as multiple rolls of coins arranged in an even manner inside the stroma (figure 1*b*) [15].

During the next years, the architecture of the lamellar system inside the chloroplast stayed in focus. The application of electron microscopy further facilitated elucidation of the chloroplast structure. Initial attempts to visualize chloroplasts were conducted with tobacco [63]. A first electron microscopic cross-section through a tulip chloroplast is shown in

figure 1*a* [14]. It could furthermore be shown that the previously described granules consisted of stroma lamellae and grana lamellae, which made it clear that the grana do not occur isolated inside the organelle but rather are interconnected via stroma lamellae resulting in a complex network [18]. In the 1960s, the structural units of this lamellar system were named thylakoids for the first time, referring to the Greek word 'ϑνλακοειδής' meaning 'sack-like' [20]. Stroma and grana thylakoids were thought to alternate randomly in the lamellar system. Edges of the stacked grana

royalsocietypublishing.org/journal/rsob    Open Biol. 9: 180237

regions that are in contact with the stroma were named margins, while the outermost thylakoid membrane was described as the end-granal membrane [41].

At the same time, focus was also laid on thylakoid structure and composition, as it was already speculated to be the place of the light-dependent reactions of photosynthesis [64]. The lamellar system of mature chloroplasts was estimated to contain 8% chlorophyll and 40% lipids. Moreover, it was stated that protein and lipids would lie side by side in separate layers within the thylakoid membrane and possibly interacting. In this model, the lipid layer would be topped by an additional globular protein sheet. Still, it was controversially discussed whether the thylakoid membrane was symmetric or asymmetric concerning its composition, and how proteins and lipids would spatially relate to each other [41,65]. Later on, data from X-ray diffraction and electron microscopy were combined, which subsequently led to the idea of the thylakoids being a bilayer interrupted with a large number of proteins referred to as 'particulate elements' [66]. But knowledge of the molecular organization of the thylakoid membrane remained limited. As a structural heterogeneity in the form of stroma lamellae and grana stacks was apparent, it was tempting to also suggest a functional heterogeneity. Evidence for this purpose followed soon as it could be shown that non-appressed regions of the thylakoids were highly enriched in PSI as well as ATPase, whereas grana partitions mainly possessed PSII with its corresponding light-harvesting complexes (LHCII) [67,68]. Today, it is known that the thylakoid lipid bilayer shows a unique composition. It is estimated to contain about 30% lipids of which 70–80% are galactosyl diglycerides in the form of monogalactosyl diacylglycerol (MGDG) and digalactosyl diacylglycerol (DGDG). Interestingly, MGDG as the main galactolipid of the thylakoids is a non-bilayer forming lipid due to its two highly unsaturated fatty acyl chains. In addition to galactosyl diacylglycerides, the thylakoid membrane also contains around 10% sulfoquinovosyl diacylglycerol and small amounts of phosphatidylglycerol and phosphatidylcholine. These lipids are thought to be unevenly distributed between the stromal and the luminal leaflet of the thylakoids [8,69]. As previously assumed, the same is true for the dominant protein complexes of the thylakoids, which are PSI [70] and PSII [71], with their respective associated light-harvesting antenna, the cytochrome $b_6f$ complex [72] and the chloroplastic ATP synthase [73]. While PSII is more prominent in the grana stacks, PSI and ATP synthase dominate in the stroma lamellae [74].

Moreover, focus was set on the three-dimensional structure of the thylakoid system. In this context, extensive sectional series through grana were made and it was stated that two types of stacking existed. These could be distinguished as either 'disjunctive' meaning spatially or locally separated thylakoids or 'conjunctive' referring to coherent thylakoids joined in a serial manner. With that, the luminal space was initially not believed to be a continuous system [23,75].

Different models were proposed for the unique architecture of the thylakoid membrane. In 1970, Paolillo [76] suggested the helical fretwork model in which the stroma lamellae, here referred to as frets, are helically arranged around the cylindrical grana stacks. In contrast to previous viewpoints, this model stated the continuity of the thylakoids as multiple frets that were thought to wind around each granum creating a highly interconnected fretwork. Furthermore, it was found that all present helices are wound around the grana stacks as multiple,

right-handed helices [76]. Junctional connections between the spiralling frets and the grana thylakoids appear as slits in the grana margins and can vary in size. This led to the assumption that junctional slits may participate in the functional regulation of thylakoid activities [77]. Experiments that used three-dimensional reconstruction to gain insights into the formation and structure of the thylakoids also considered the grana membranes to be associated with stroma thylakoids in a helical way [42].

An alternative model known as the forked membrane model was suggested. Here, a single continuous membrane is thought to be at the origin of thylakoid formation. The complexity of grana regions is gained through folding and stacking of bifurcated stroma lamellae which fuse within the granum body [67,78]. With that, grana are built of repeating units that are rotated in a relative manner to each other around the axis of the granal cylinder. Also here, all parts are directly connected to neighbouring membranes resulting again in an interconnected morphology [79]. A current view on chloroplast ultrastructure and internal organization is given in figure 1e,f.

It was stated for a long time that lateral heterogeneity of photosynthetic protein complexes as well as the three-dimensional structure of the thylakoids essentially contributed to a balanced distribution of excitation energy. Now, it is rather assumed that redox balance is mainly achieved by reversible phosphorylation of PSII–LHCII supercomplexes, by slowing down the electron transport and by thermal dissipation of excess energy [80]. Nevertheless, the unique and complex intertwined structure of the thylakoid membrane is strongly interconnected with its proper function. Many gaps remain still to be filled when it comes to elucidating the three-dimensional structure of this unique membrane system.

# 4. The big mystery of thylakoid biogenesis

Besides its structure, the exact mechanisms by which the thylakoid membrane itself is formed largely remain elusive to date. In general, thylakoids are very dynamic as they must adapt rapidly to environmental changes and stresses by changing their lipid and protein content [81]. But surprisingly, little is known about how and where the numerous protein subunits as well as hundreds of cofactors are assembled to eventually build functional complexes during thylakoid biogenesis.

In cyanobacteria and green algae, there is evidence for specialized membranous compartments involved in the synthesis and assembly of photosynthetic compartments. In the cyanobacterium *Synechocystis*, so-called PratA-defined membranes (PDMs) were identified as distinct regions at zones where thylakoids and plasma membrane converge. PratA has been described as a tetratricopeptide repeat protein responsible for the binding and delivery of $Mn^{2+}$ ions to PSII pre-complexes. It is assumed that PDMs resemble biogenesis centres that function as nucleation points where PSII proteins are preloaded with $Mn^{2+}$ while the final assembly is accomplished after transfer to the developing thylakoid lamellae (figure 2a) [50,82].

As chloroplasts started as primary endosymbionts including a massive reorganization of gene regulation and coordination, thylakoid biogenesis in plastid-containing organisms is logistically more complex than in cyanobacteria. Green algae like *Chlamydomonas reinhardtii* contain only one

single chloroplast with concentric thylakoids. Inside this chloroplast, a subcellular microcompartment called the pyrenoid helps with fixing $CO_2$. Around the pyrenoid, a specific cytological region named 'translation (T)-zone' was detected where PSII subunit-encoding mRNAs and ribosomes co-localize in distinct foci. The T-zone is therefore believed to also represent a specialized localization of PSII subunit synthesis and assembly (figure 2a) [50,51].

Chloroplasts of land plants contain a more complex and intertwined thylakoid network (figure 2c,e). It is known that many of the needed components of the thylakoid membrane such as lipids or pigments originate from the inner membrane [8,83]; especially galactolipids like MGDG and DGDG are essential for thylakoid formation [84]. Both lipids are produced at the envelope membranes. DGDG assembly takes place in the outer envelope while MGDG is assembled in the inner envelope, where also its main producing synthase MGD 1 is located. Considering that the inner envelope produces lipids for the thylakoids, it is not surprising that both membranes share a similar lipid composition [85]. Furthermore, it was hypothesized that the apoproteins of LHCII bind to pigments in the inner envelope. This process is thought to stabilize initial intermediates and to promote assembly of LHCs within the chloroplast envelope [86,87]. With that, the question arises how these hydrophobic components bridge the aqueous stroma in order to reach the thylakoid membrane. In principle, there are three possible ways that lipids, proteins and small organic molecules could be trafficked from the inner plastid envelope membrane to the thylakoids (figure 2d) [10,52].

First, components could reach their target through stable connections that would form lateral fusions between the two membranes. As already shown, invaginations of the inner envelope have been observed very early and were thought to contribute to the formation of PTs in differentiating proplastids (figure 2e) [11]. Even though sometimes a continuum between the inner envelope and the developing internal membrane system in the early stages of plastid differentiation can be observed, mature chloroplasts as well as cyanobacteria show no connection between the inner envelope or the plasma membrane and the thylakoids, respectively [52]. While early formation of thylakoids might be achieved by invaginations of the inner envelope, this is unlikely to happen in later stages [8].

Second, soluble transfer proteins inside the stroma could function as a shuttle for thylakoid building blocks. So far, there is no clear evidence for the existence of such transfer proteins inside chloroplasts as no putative proteins can be identified as yet [88].

A third idea assumes the existence of a vesicle transport system inside the chloroplast. The suggested vesicle transfer could be similar to the ones observed in the cytosol such as the secretory pathway, endocytosis, neural transmission and vacuole formation [9].

# 5. Plastid vesicle transport

Even though for a long time it was believed that initial thylakoids in proplastids were produced from invaginations of the inner membrane of the plastid envelope [89], vesicular structures in the sense of round-shaped globules have often been detected inside young chloroplasts. In early papers, observed alleged vesicles were distinguished as different types named 'osmiophilic droplets' [90] or 'inclusion bodies' of the proplastid [91]. In today's definition, vesicles are large structures with a lipid bilayer that enclose an interior space.

Moreover, it was already known that vesicles would accumulate in the stroma after exposure to stress [91] or low temperature treatment [92]. Upon incubation at $12°C$, it could be shown in pea that vesicles occurred within the stroma between the inner envelope and the thylakoid membrane. The formation of these temperature-dependent vesicles had previously also been observed in the cytosol of animal cells [93]. Based on this, it was estimated that vesicle fusion with the target membrane was hindered by cold treatment in both cases. Simultaneously with the appearance of stromal vesicles, also invaginations that were continuous with the inner envelope could be captured in electron microscopy [92]. Furthermore, it was shown that the use of established eukaryotic inhibitors and competitors such as protein phosphatase inhibitors, caldmodulin inhibitors or calcium antagonists would also lead to the accumulation of stromal vesicles by preventing fusion. Hence, vesicle formation at the donor membrane could be inhibited by a non-hydrolysable GTP, which led to the assumption that budding could be controlled by a GTPase similar to the eukaryotic secretory pathway of the endosomal vesicle trafficking system [94].

Another vesicular phenomenon that can especially be observed in C4 plants is the peripheral or plastid reticulum (PR) that is found in chloroplasts [95]. Early on, the PR was described as tubular double membranes continuous with the inner envelope as well as the thylakoids, putatively facilitating the movement of materials from one membrane to another [25]. Between the inner envelope and the thylakoid membrane, the PR appears as a dense layer of vesicles. Whether a PR also exists in C3 plants is currently debated. Its physiological role is still completely unknown.

Vesicles have most often been observed either in proplastids or in developing chloroplasts (figure 2e). Their presence can be increased in chloroplasts under certain circumstances such as low temperature or special pre-treatment. Either way, it could be shown that vesicles are a persistent feature not only of chloroplasts but also of other forms of plastids such as etioplasts, leucoplasts, chromoplasts and desiccoplasts. This is true for plastids of C3 and C4 plants, different cell types and different organs. Observed vesicles were generally 50 nm in diameter and occurred with an approximate frequency of 1–8 vesicles per plastid section [96]. A possible reason for the rare detection of vesicles in mature chloroplasts could be their velocity. It was shown that diffusion velocities of molecules differ between stroma, cytosol and aqueous solution. Furthermore, two-photon excitation fluorescence correlation spectroscopy revealed that GFP units within stromal-filled tubules (stromules) actively moved with a velocity of about $0.12 \, \mu m \, s^{-1}$ [97]. Even though stromules differ from stromal vesicles as they are not present inside the organelle but form tubular connections between plastids, stromules give a first hint on how fast movement through the stroma could be.

Besides, no such vesicle system could be found in cyanobacteria [94]. Taking this into account, it is very likely that the chloroplast vesicle system is of eukaryotic origin and that the cytosolic system was transferred into the organelle. Evolutionarily, it could be shown that such a trafficking system does not exist in any lineage of algae. It was exclusively

royalsocietypublishing.org/journal/rsob    Open Biol. 9: 180237

royalsocietypublishing.org/journal/rsob Open Biol. 9: 180237

found in organisms belonging to the embryophytes comprising bryophytes (mosses and liverworts), pteridophytes (ferns, horsetails and lycophytes) and spermatophytes (seed plants). Interestingly, embryophytes evolved simultaneously with the challenging transition to a life outside water, which correlated with the development of extensive tissue organization as well as a complex thylakoid structure. The colonization of this new habitat most probably required enormous adaptation and maintenance of the photosynthetic apparatus and might therefore be one of the reasons for the establishment of a vesicle transport system [98]. But as detailed proof of involved proteins is lacking, it cannot be excluded that also prokaryotic traits are important for plastid vesicle transport. The same has been described for the import machinery found in the envelope membranes. These hetero-oligomeric protein complexes consist of both eukaryotic and prokaryotic derived components [99]. Taking this into account, it is conceivable that the eukaryotic system was acquired by plastids in the first instance to then be transformed into a unique and new trafficking system by altering and adapting the function of prokaryotic proteins.

As biochemical and molecular biological evidence for a vesicular transport system inside chloroplasts increased, the search for homologous components of the cytosolic vesicle system started. The secretory pathway in plants consists of the nuclear envelope, the endoplasmic reticulum (ER), the Golgi apparatus and various post-Golgi intermediate compartments, the vacuoles, the lysosomes and the plasma membrane. Between all these individual compartments, many steps are cyclic and therefore comprise anterograde and retrograde transport [100]. The initial vesicle budding can be mediated by three different kinds of coat protein complexes (COPs), namely COPI, COPII and clathrin. These coats are supramolecular assemblies of proteins that initiate membrane deformation and participate in cargo selection. Clathrin coats are needed in post-Golgi locations, whereas COPI is involved in intra-Golgi transport and retrograde transport from the Golgi apparatus to the ER. COPII protein coats mediate the export from the ER to the Golgi complex [101]. Regulation of the coat proteins is achieved by small GTPases such as the ADP ribosylation factor (ARF) for COPI-coated vesicles or the secretion-associated RAS-related protein 1 (Sar1) for COPII-coated vesicles. Fission of clathrin-coated vesicles requires the large GTPase dynamin. Vesicle targeting and fusion is mediated by membrane-spanning soluble $N$-ethylmaleimide-sensitive factor attachment receptor (SNARE) proteins and regulated by RAB GTPases. SNAREs are located on the vesicle membrane (v-SNAREs) as well as on the target membrane (t-SNAREs) where they support the fusion process and the delivery of the cargo protein by forming a *trans*- and a *cis*-complex. For some of these components and additional associated factors, homologues targeted to the chloroplast could be identified by bioinformatics approaches. Most of the identified homologues corresponded to yeast COPII-coated vesicle components [102–104]. So far, only two of the identified candidates, namely chloroplast-localized SAR1 (cpSAR1) and chloroplast-localized RabA5e (cpRabA5e), were experimentally confirmed to be chloroplast proteins. cpSAR1 was verified to have GTPase activity and to be dually located in the inner envelope as well as in chloroplast vesicles [105]. Nevertheless, it is debated whether cpSar1 really is a homologue of the yeast Arf GTPase Sar1p, which belongs to the extended Ras-like family, or whether it rather belongs to the OBG-like GTPases,

a subfamily of bacterial P-loop GTPases. In prokaryotes, OBG-like proteins play an important role during sporulation and differentiation. Homologues of these GTPases are widely distributed among algae and plants, indicating that the protein has retained an important function even though functional adaptation has to be considered. In this context, cpSar1 was renamed AtOBGL and suggested to be essential for embryo development in *Arabidosis* [106]. This proposal contrasts with a possible participation in plastid vesicle transport. cpRab5Ae was found to be localized in the stroma and the thylakoids. Owing to its homology to yeast Ypt31/32, it is also thought to have a role for regulating vesicle transport [107]. Nevertheless, it must be emphasized that the final proof of a role in vesicle transport is still missing for both proteins.

In addition to bioinformatic approaches, numerous mutant studies have been conducted to unravel putative proteins involved in thylakoid biogenesis as well as plastid vesicle transport.

One protein with a critical role in thylakoid biogenesis is Vipp1 (vesicle-inducing protein in plastid 1). It was first described in pea as the inner membrane-associated protein of 30 kDa (IM30) [108]. Vipp1 derived from the bacterial phage shock protein A (PspA) via gene duplication. During duplication of the *psbA* gene, a novel α-helical extension at the C-terminal domain comprising 30 amino acids was added that seems essential for its function in thylakoid formation. As Vipp1 is exclusively found in cyanobacteria and chloroplasts, it probably developed in parallel to the emergence of the thylakoid membrane system.

In contrast to algae and higher plants, cyanobacteria still possess both proteins, PspA and Vipp1, which strengthens the hypothesis that algal and plant Vipp1 evolved from a cyanobacterial PspA [109–112]. Furthermore, it could be shown that PspA in cyanobacteria carrying a mutation in the *vipp1* gene is not sufficient to compensate the deficiency [109,110], while cyanobacterial as well as plant Vipp1 can functionally complement *Escherichia coli* PspA [113,114]. Interestingly, the unicellular photosynthetic cyanobacterium *Gloeobacter violaceus* sp. PCC 7421 lacks both Vipp1 and a thylakoid membrane system. The only present membrane unit is the cytoplasmic membrane where photosynthesis is thought to take place in specialized lipid domains [115,116]. The example of *Gloeobacter* is another strong indication that Vipp1 and thylakoid biogenesis are linked.

More recently, strong focus was laid on investigating the detailed physiological function of Vipp1 in generating and/or maintaining the thylakoid membrane system. It was found in cyanobacteria that Vipp1 forms oligomeric rings that disassemble to located puncta. These puncta for their part associate with highly curved regions of the thylakoid membrane where they are supposed to act as scaffold-like aggregates. Taken together, it is very likely that Vipp1 fulfils a protective function for membranes with high curvature stress in order to control integrity and biogenesis of thylakoid membranes [117,118]. Membrane remodelling, in general, requires nucleotide binding and/or hydrolysis. Indeed, it was shown that Vipp1 as well as its homologue PspA are capable of both GTP binding and hydrolysis. Even though Vipp1 does not contain a canonical G domain that is typical for GTPases, heterologously expressed Vipp1 seems to bind and hydrolyze GTP via its N-terminal α-helix *in vitro*, which seems to promote oligomerization. With that, Vipp1 could represent a novel type of GTPase acting in chloroplast membrane fusion and/or remodelling [119,120].

royalsocietypublishing.org/journal/rsob    Open Biol. 9: 180237

Further suggested players have been reviewed elsewhere [88]. Besides Vipp 1, most promising are the fuzzy-onion (FZO)-like protein (FZL) [121,122], the thylakoid formation protein 1 (THF1) [123] and snowy cotyledon 2 (SCO2) [124,125]. All of them show interesting phenotypes that indicate a role in thylakoid biogenesis and vesicle transport. Loss of function mutants of FZL and Vipp1 show an altered thylakoid structure. Vipp1-depleted plants furthermore display fewer vesicles and a disturbed photosynthetic electron transport chain. By contrast, more vesicles accumulate in the stroma of *fzl*, *thf1* and *sco2*. SCO2 mutants also show impaired chloroplast development at the cotyledon stage, but this normalizes later and leads to green leaves. Besides that, SCO2 and THF1 both have been shown to interact with LHCB proteins [126,127], which have been suggested to be a possible cargo in vesicles [103]. Despite numerous promising experiments to elucidate vesicle transport and thylakoid biogenesis, it must be underlined that none of the proteins presented plays a proven role in these processes. The exact function of the individual proteins for the chloroplast must be further investigated. Until then, all assumptions remain speculative.

Altogether, it can be presumed that vesicles exist in chloroplasts at different developmental stages as well as in different forms. In general, it can be said that vesicles occur more often in young stages of development. This suggests that the transport of vesicles plays an important role particularly during the early biogenesis of thylakoids. Vesicles could not only be observed upon specific pre-treatments, but also occur in natural plastids. With that it is excluded that these vesicles are an artefact of electron microscopic preparation techniques. Yet their general function and contribution to thylakoid biogenesis is largely unknown. Bioinformatic approaches suggesting similarity to the COPII vesicular trafficking system of the cytosol made it tempting to also assume plastid vesicles as cargo shuttles for thylakoid building blocks. This plastid vesicle system could be an ongoing and protein-mediated transport in order to build and maintain the thylakoid network. An overview of putative processes involved in thylakoid biogenesis is depicted in figure 3. Nevertheless, exact processes and involvements remain an open question.

# 6. Thylakoid revival—resurrection plants as a new model system?

So far, the issue of thylakoid biogenesis has been addressed by several different approaches described in this review. Besides, a putative new model system could provide a different view on thylakoid biogenesis and maintenance. Desiccation-tolerant or resurrection plants are stunning organisms that can manage and survive times of extreme dryness by having evolved a set of unique strategies. Resurrection plants can tolerate up to 95% of total cellular water loss and are furthermore able to fully recover upon rehydration [131]. The major problem of water deficiency lies in the enduring presence of light. As many species of resurrection plants are found in the southern parts of Africa [132], continuous high light conditions lead to overexcitation of the photosynthetic apparatus. As stomata are closed under water-deficit stress, the electron transport chain gets over-reduced that eventually leads to the transfer of electrons back to water which in turn produces toxic reactive oxygen species [133]. Angiosperm resurrection plants have developed mainly two different

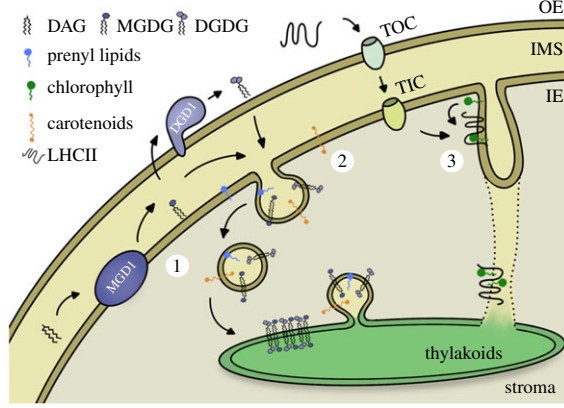

**Figure 3.** Suggestion of processes contributing to thylakoid biogenesis in chloroplasts of land plants. Lipids, prenyl lipids, pigments and proteins are needed to build a functional thylakoid membrane. These components must somehow travel through the aqueous stroma to reach the thylakoids. 1. The galactolipids DGDG and MGDG are synthesized from their synthases DGD1 and MGD1 at the outer and inner envelope, respectively [69,85,128]. 2. Together with prenyl lipids [129] and carotenoids [130] that are also made at the inner envelope, they could be a putative cargo of a suggested plastid vesicle transport system. Vesicles are thought to bud off from the inner envelope, travel across the stroma to finally fuse with the thylakoid membrane to deliver their cargo. 3. Another connection between both membranes could be provided by invaginations forming lateral fusions. Plastid-targeted proteins like LHCII are imported via the TOC and TIC translocons residing in the envelope membranes. These proteins could furthermore be pre-assembled with chlorophylls [86,87] in the incipient invaginating membrane before being delivered to their target membrane.

solutions to this problem. Based on the used strategy to shut down photosynthesis during desiccation, resurrection plants are classified in either homoiochlorophyllous species or poikilochlorophyllous species. Homoiochlorophyllous resurrection plants retain their chlorophyll but achieve protection through pigment production ('sun screen' pigments like carotenoids and anthocyanins) and several morphological changes such as leaf folding. On the other hand, poikilochlorophyllous res urrection plants (PDTs) use an even more extreme protection mechanism. During drying, they totally dismantle their thylakoid membranes and break down all chlorophyll [131,134]. Furthermore, these plants also downregulate genes encoding several subunits of PSII such as *psbR*, *psbA* or *psbP* [135]. At the time water is available again, photochemical activity recovers, which requires a rapid reconstruction of the thylakoid membrane as well as the photosynthetic apparatus. How this process is fulfilled upon rehydration remains elusive, but it strikingly resembles the processes that occur during photomorphogenesis from etioplasts to chloroplasts [136].

*Xerophyta* is a monocotyledonous plant genus that encompasses several PDTs including *Xerophyta viscosa* and *Xerophyta humilis*. As chloroplasts of these plants totally lose their typical internal thylakoid network upon extreme dryness, they turn into a unique form of plastid called the desiccoplast [131] or xeroplast [136]. Desiccoplasts still hold an envelope, but all thylakoids are broken down to small vesicles. In addition, circular membranous structures occur that are thought to derive from invaginations of the boundary membrane. The fate of these structures is not further known. With that, the inside of a desiccoplast differs strikingly from that of a chloroplast as only vesicles, membrane circles and plastoglobuli are present [131,134]. Upon

rehydration, several distinct stages in the desiccoplast–chloroplast transition were identified. The rebuilding of intact and active thylakoid membranes starts with the elongation of membrane-attached vesicles forming PTs. As the plastid extends, starch granules accumulate and thylakoid precursor membranes are formed from the PTs as a first step of the desiccoplast–chloroplast transition. As the disappearance of membrane-attached vesicles and the formation of PT membranes clearly correlates, it can be concluded that the revived thylakoids may form with little contribution from the inner envelope. It is further believed that the stockpile of MGDG and DGDG in membrane-attached vesicles in desiccoplasts is sufficient for thylakoid assembly. Beside these differences to processes in the proplastid, there is strong evidence that the molecular mechanisms underlying the transition from etioplasts or desiccoplasts to chloroplasts are similar [134]. The final stacking of thylakoid membranes as well as the syntheses of the PSII core protein D1, the chlorophyll-binding protein Lhcb2 and the DGDG synthase 1 are again light-dependent, as already has been described for the etioplast–chloroplast transition [136]. If light is absent, desiccoplasts turn into etioplasts containing PTs and PLBs. When the plant then gets illuminated, these desiccoplast-derived etioplasts transform into regular chloroplasts within 3 days [137]. This is much slower than in normal land plants where the process often only takes a few hours.

Desiccation tolerance is common in seeds, spores, pollen grains and various other organisms. Based on novel omics technologies, it was suggested that vegetative desiccation tolerance in resurrection species arose by redirection of genetic information from seeds [138,139]. Still, many question concerning exact molecular processes and regulation pathways remain unanswered to date.

# 7. Conclusion

Numerous pioneering experiments lead from the first insights into thylakoid morphology in the middle of the nineteenth century to modern views on chloroplast differentiation, thylakoid biogenesis and structure. Stunning breakthroughs were gained in all mentioned fields. Today, we know that chloroplasts derived from proplastids present in the plant stem cells. These progenitors can either directly transform into mature chloroplasts upon illumination or turn into etioplasts under the absence of light. Beside these stages, many other plastid forms exist depending on the functional demand of the corresponding tissue. Developmental gradients towards mature chloroplasts exist not only in monocots, but also in meristematic cells as well as leaves of dicots. Here, different stages of development were not only found between leaves of different ages, but also within a leaf.

Chloroplasts furthermore contain a complex and intertwined membrane network called the thylakoids. Very early it was known that this network consists of stacked regions named the grana thylakoids that are interconnected by non-appressed sections, the stroma thylakoids. The thylakoid membrane is characterized by a unique composition of proteins, lipids, pigments and multiple cofactors. As MGDG, one of the two major lipids, is a non-bilayer forming lipid, the interplay of lipids and proteins seems to be important for thylakoid formation. Another feature is the lateral heterogeneity of the major photosynthetic protein complexes as they are asymmetrically distributed among the thylakoid membrane. To date, the three-dimensional structure is still debated, but a helical model is favoured.

Beside all structural and morphological insights, very little is known about mechanisms for thylakoid biogenesis. As many important building blocks derive from the inner envelope, they could either bridge the aqueous stroma via invaginations, soluble transfer proteins or a plastid vesicle transport. The latter has been in strong focus for the last years as it was proved that vesicles are a persistent feature of plastids and that they accumulate in the stroma under special conditions. As vesicle trafficking is a common feature of the secretory pathway in the cytosol, it was hypothesized that the putative plastid vesicle transport system could be of eukaryotic origin. This idea is further supported by the lack of such a system in cyanobacteria as well as in algae. Since the first observations of vesicles, many proteins have been suggested as putative players. But still, details on molecular processes and regulators remain elusive.

An interesting new model system to investigate thylakoid biogenesis is resurrection plants. These plants grow in areas of extreme dryness and manage to survive up to 95% of cellular water loss. Evolution of different strategies led to solutions to deal with the problem of water deficiency combined with overexcitation through constant light. One of these solutions is the reversible breakdown of chlorophyll and dismantling of the thylakoid membrane into vesicles. During this process, the chloroplasts turn into so-called desiccoplasts. Upon rehydration, the desiccoplast is able to revive the thylakoid membrane including functional photosynthesis.

This review summarized findings on thylakoid biogenesis and pointed to new possible ways to approach this complex topic. Furthermore, it made clear that many gaps still need to be filled experimentally. Considering this, thylakoid biogenesis is definitely not a past-time topic, but rather remains a stunning field to explore for the future.

Data accessibility. This article has no additional data.

Authors' contributions. A.M. wrote the manuscript. S.S. revised the manuscript and prepared the figures, and J.S. supervised the study and revised the manuscript.

Competing interests. We declare we have no competing interests.

Funding. A.M. was awarded a FCI Kekulé scholarship. This work was supported by the Deutsche Forschungsgemeinschaft (DFG), SFB-TR 175, projects B05 to J.S. and B06 to S.S.

Acknowledgements. We thank Christel Glockmann, University of Kiel, for excellent technical assistance with the preparation of TEM pictures.

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
