## [Reviewer comments · Open Biology]

Review History

RSOB-18-0237.R0 (Original submission)

Review form: Reviewer 1

Recommendation

Accept with minor revision (please list in comments)

Are each of the following suitable for general readers?

- a) **Title**
Yes
- b) **Summary**
Yes
- c) **Introduction**
Yes

Is the length of the paper justified?

Yes

Should the paper be seen by a specialist statistical reviewer?

No

Is it clear how to make all supporting data available?

Yes

Is the supplementary material necessary; and if so is it adequate and clear?

Not Applicable

Do you have any ethical concerns with this paper?

No

Comments to the Author

Manuscript titled: "Brief history of thylakoid biogenesis by Irene Annabel Mechela, Serena Schwenkert, and Jürgen Soll is a review on research on the thylakoid membrane biogenesis and its structure, both in evolutionary and historical aspects.

- 1) "The introduction" gives an overall information on the importance of photosynthesis and on the endosymbiotic origin of chloroplasts.
- 2) "The Chloroplast differentiation - in the beginning, is the proplastid" is an overview of research dealing with differentiation of chloroplasts from proplastids or through the stage of etioplasts to mature chloroplasts, in historical aspects of discoveries till the recent times. This part is interestingly written; especially valuable is the description of the connection of chloroplast photomorphogenesis with the leaf primordium development. This manuscript also takes into account a recent important result of Gügel and Soll, 2017, that chloroplast differentiation exhibits a developmental gradient along the leaf blade not only in monocotyledons but also in dicotyledons.
- 3) "From rolls of coins to reality - unraveling thylakoid morphology". In this part the authors, present, from the historical perspective, investigations dealing with the thylakoid system structure, composition, and spatial arrangement. It is important that the authors connect the thylakoid structure with membrane composition.
- 4) "The big mystery of thylakoid biogenesis" presents an evolutionary aspect of the formation of advanced and specialized membrane compartments from cyanobacteria, through green algae to land plants. The interesting point is the existence of three possible ways of connection and transfer of proteins between inner the plastid envelope and thylakoid.
- 5) "Plastid vesicle transport". This paragraph is extensive. I consider this as a good element because it has not been described in detail elsewhere. It is emphasized that plastid vesicle transport is probably of eukaryotic origin, since it is not found in cyanobacteria, and was acquired by plastids as a new trafficking system for prokaryotic proteins. Another important issue is pointing the putative proteins which probably are involved in plastid vesicle transport.
- 6) "Thylakoid revival - resurrection plants as a new model system?" An interesting paragraph giving examples of different thylakoid revivals after water shortage leading to reconstruction of the thylakoid membrane structure and of the photosynthetic function. Perhaps it would be worth to mention whether such a mechanism can be involved after other stresses.
- 7) "Conclusion" - is an interesting summary, emphasizing the main points of this review.

A problematic point is the order of citation of Figures. There is no reason why Figure 1d and then Figure 1 c cannot be placed earlier, as Figure 1 a and mentioned earlier in the text.

Is Figure 2 e performed by authors? Otherwise, the source of these TEM pictures should be given.

It would be better to give figure captions in a more detailed way, e.g. Figure 2 d.
Figure 3 - the word "Detailed" is not justified.

Concluding:

The subject of the manuscript is interesting and important for plant biologist, it deals with molecular, structural and evolutionary aspects of the thylakoid system biogenesis. The manuscript is clearly written. It is interesting and understandable also for non-specialists. After corrections mentioned above, the manuscript can be published in Open Biology.

Review form: Reviewer 2

Recommendation

Major revision is needed (please make suggestions in comments)

Are each of the following suitable for general readers?

- a) **Title**
Yes
- b) **Summary**
Yes
- c) **Introduction**
Yes

Is the length of the paper justified?

Yes

Should the paper be seen by a specialist statistical reviewer?

No

Is it clear how to make all supporting data available?

Not Applicable

Is the supplementary material necessary; and if so is it adequate and clear?

Not Applicable

Do you have any ethical concerns with this paper?

No

Comments to the Author

Please see attached.

Decision letter (RSOB-18-0237.R0)

02-Jan-2019

Dear Dr Schwenkert

We are pleased to inform you that your manuscript RSOB-18-0237 entitled "A brief history of thylakoid biogenesis" has been accepted by the Editor for publication in Open Biology. The reviewer(s) have recommended publication, but also suggest some minor revisions to your manuscript. Therefore, we invite you to respond to the reviewer(s)' comments and revise your manuscript.

Please submit the revised version of your manuscript within 14 days. If you do not think you will be able to meet this date please let us know immediately and we can extend this deadline for you.

- 1) A text file of the manuscript (doc, txt, rtf or tex), including the references, tables (including captions) and figure captions. Please remove any tracked changes from the text before submission. PDF files are not an accepted format for the "Main Document".
- 2) A separate electronic file of each figure (tiff, EPS or print-quality PDF preferred). The format should be produced directly from original creation package, or original software format. Please note that PowerPoint files are not accepted.
- 3) Electronic supplementary material: this should be contained in a separate file from the main text and meet our ESM criteria (see <http://royalsocietypublishing.org/instructions-authors#question5>). All supplementary materials accompanying an accepted article will be treated as in their final form. They will be published alongside the paper on the journal website and posted on the online figshare repository. Files on figshare will be made available approximately one week before the accompanying article so that the supplementary material can be attributed a unique DOI.

Online supplementary material will also carry the title and description provided during submission, so please ensure these are accurate and informative. Note that the Royal Society will not edit or typeset supplementary material and it will be hosted as provided. Please ensure that

the supplementary material includes the paper details (authors, title, journal name, article DOI). Your article DOI will be 10.1098/rsob.2016[*last 4 digits of e.g. 10.1098/rsob.20160049*].

4) A media summary: a short non-technical summary (up to 100 words) of the key findings/importance of your manuscript. Please try to write in simple English, avoid jargon, explain the importance of the topic, outline the main implications and describe why this topic is newsworthy.

Images

Data-Sharing

It is a condition of publication that data supporting your paper are made available. Data should be made available either in the electronic supplementary material or through an appropriate repository. Details of how to access data should be included in your paper. Please see <http://royalsocietypublishing.org/site/authors/policy.xhtml#question6> for more details.

Data accessibility section

Sincerely,

The Open Biology Team
<mailto:openbiology@royalsociety.org>

ditage Insights by clicking on the following link: <https://www.surveymonkey.com/r/author-perspectives-on-academic-publishing-royal-society>

This should take no more than 15 minutes and you will have the opportunity to enter a prize draw. We hope these results will provide us with valuable insights we can use to improve our service.

Reviewer(s)' Comments to Author:

Referee: 1

Comments to the Author(s)

Manuscript titled: "Brief history of thylakoid biogenesis by Irene Annabel Mechela, Serena Schwenkert, and Jürgen Soll is a review on research on the thylakoid membrane biogenesis and its structure, both in evolutionary and historical aspects.

- 1) "The introduction" gives an overall information on the importance of photosynthesis and on the endosymbiotic origin of chloroplasts.
- 2) "The Chloroplast differentiation - in the beginning, is the proplastid" is an overview of research dealing with differentiation of chloroplasts from proplastids or through the stage of etioplasts to mature chloroplasts, in historical aspects of discoveries till the recent times. This part is interestingly written; especially valuable is the description of the connection of chloroplast photomorphogenesis with the leaf primordium development. This manuscript also takes into account a recent important result of Gügel and Soll, 2017, that chloroplast differentiation exhibits a developmental gradient along the leaf blade not only in monocotyledons but also in dicotyledons.
- 3) "From rolls of coins to reality – unraveling thylakoid morphology". In this part the authors, present, from the historical perspective, investigations dealing with the thylakoid system structure, composition, and spatial arrangement. It is important that the authors connect the thylakoid structure with membrane composition.
- 4) "The big mystery of thylakoid biogenesis" presents an evolutionary aspect of the formation of advanced and specialized membrane compartments from cyanobacteria, through green algae to land plants. The interesting point is the existence of three possible ways of connection and transfer of proteins between inner the plastid envelope and thylakoid.
- 5) "Plastid vesicle transport". This paragraph is extensive. I consider this as a good element because it has not been described in detail elsewhere. It is emphasized that plastid vesicle transport is probably of eukaryotic origin, since it is not found in cyanobacteria, and was acquired by plastids as a new trafficking system for prokaryotic proteins. Another important issue is pointing the putative proteins which probably are involved in plastid vesicle transport.
- 6) "Thylakoid revival – resurrection plants as a new model system?" An interesting paragraph giving examples of different thylakoid revivals after water shortage leading to reconstruction of the thylakoid membrane structure and of the photosynthetic function. Perhaps it would be worth to mention whether such a mechanism can be involved after other stresses.
- 7) "Conclusion" - is an interesting summary, emphasizing the main points of this review.

A problematic point is the order of citation of Figures. There is no reason why Figure 1d and then Figure 1 c cannot be placed earlier, as Figure 1 a and mentioned earlier in the text. Is Figure 2 e performed by authors? Otherwise, the source of these TEM pictures should be given. It would be better to give figure captions in a more detailed way, e.g. Figure 2 d. Figure 3 - the word "Detailed" is not justified.

Concluding:

The subject of the manuscript is interesting and important for plant biologist, it deals with molecular, structural and evolutionary aspects of the thylakoid system biogenesis. The manuscript is clearly written. It is interesting and understandable also for non-specialists. After corrections mentioned above, the manuscript can be published in Open Biology.

Referee: 2

Comments to the Author(s)
Please see attached.

Author's Response to Decision Letter for (RSOB-18-0237.R0)

See Appendix A.

Decision letter (RSOB-18-0237.R1)

09-Jan-2019

Dear Dr Schwenkert

We are pleased to inform you that your manuscript entitled "A brief history of thylakoid biogenesis" has been accepted by the Editor for publication in Open Biology.

Sincerely,

The Open Biology Team
mailto: openbiology@royalsociety.org

Appendix A

Reviewer 1

"Thylakoid revival - resurrection plants as a new model system?" An interesting paragraph giving examples of different thylakoid revivals after water shortage leading to reconstruction of the thylakoid membrane structure and of the photosynthetic function. Perhaps it would be worth to mention whether such a mechanism can be involved after other stresses.

So far, the desiccation is only known in response to extreme dryness. We therefore cannot comment on other stresses, which would be an interesting issue.

A problematic point is the order of citation of Figures. There is no reason why Figure 1d and then Figure 1c cannot be placed earlier, as Figure 1a and b are mentioned earlier in the text.

We preferred to keep the Figure as was, but we corrected the mentioning in the text, which is now consecutively.

Is Figure 2e performed by authors? Otherwise, the source of these TEM pictures should be given.

Yes, this was performed by the authors. This is now mentioned more explicitly in the acknowledgement.

It would be better to give figure captions in a more detailed way, e.g. Figure 2d.

A more detailed description of Figure 2d is given.

Figure 3 - the word „Detailed“ is not justified.

Was changed.

Reviewer 2

1. Page 2, lines 43-47. The sentences don't come across well. Please rewrite. I am not sure what the authors mean by words such as "in accordance" "integrated" in here.

The sentences were rephrased.

2. Page 2, line 56. "...Due to their unique composition..". This sentence is disjointed and does not convey the intended meaning.

The sentence was rephrased.

3. The readability and clarity of the section "Chloroplast differentiation – in the beginning is the proplastid" is really poor compared to "Introduction" section. I think the use of the language is really problematic here with many odd phrases and sentences.

Phrasing has been revised by the authors, see also 8./9.

4. Page 3, line 66-67. In the opening sentence "to fully mature chloroplasts" is redundant.

The sentence was rephrased.

5. Page 3, line 70. "...mostly in form of..." => "...mostly in the form of..."

Was changed.

6. Page 3, line 72. I am not sure what the authors mean by "complemented".

Was changed to 'transformed into'.

7. Page 3, line 76-77. I am not sure what is meant by "fragmentation of the stroma in meristemic tissues". Do the authors mean that the proplastid thylakoid and stroma will undergo fission to make more proplastids?

No, Strugger believed in the continuity of chloroplasts. By this, he described the presence of a primary granum in proplastids out of which "multigranular" (meaning mature) thylakoids would later on emerge. The fragmentation of the stroma described the process of starch accumulation which would increase the size of the chloroplast according to Strugger. This paragraph was rewritten.

8. On page 4, first paragraph. Words like "thickening", "splitting" and "Strugger's theory" are mentioned without any explanation of what they are. Again, on lines 91-92 and 99-101, phrases like "stated" and "considered" are used without any context or further explanation.

9. I think the authors need to better articulate the historical account of thylakoid biogenesis research. It is not clear from their writing whether some of the discussed ideas are current and relevant or discredited/unsupported.

Historical accounts were set in context to the corresponding authors of that time to ensure better guiding through the different ideas.

10. Page 4, line 113. I think “components” may be preferable to “compounds”.

Was changed.

11. Rest of the sections in the manuscript read better and there is a good discussion and synthesis of ideas on thylakoid biogenesis. At a few places though there are some vague use of words and phrases, which could be addressed in a revised manuscript.

12. Page 8, line 245. “continuous” may be more appropriate than “coherent”.

Was changed.

13. I feel that in section “plastid vesicle transport”, the VIPP1 protein is given a short shrift, given the critical role of this protein in thylakoid biogenesis. There is no discussion of the cyanobacterial homologue of Vipp1 or its absence in the non-thylakoid forming *Gloeobacter violaceus*. There is some recent interesting literature on this protein. A more detailed discussion may therefore be appropriate for this manuscript on thylakoid biogenesis.

We have added a paragraph on VIPP1, which is now found from line 427-452.

14. The authors mention that the CURT proteins are important for thylakoid biogenesis and plastid vesicle transport. I am not sure how relevant is it for these two processes. CURT proteins have been recognized for their role in creating the tight membrane curvature in grana margins and thus plants lacking these proteins have large granal diameter. Is it possible that the CURT proteins are important for the initial tabulation of prothylakoids from the inner envelope as these proteins can constrict the membrane regions.

We agree on the concerns including CURT proteins in this context and deleted it from the manuscript.

15. The authors mention that the lack of sightings of plastid vesicles may be the result of their high velocity. In support of this, the authors discuss the stromule dynamics. I am not sure how relevant is this comparison because the stromules are tubular outward projections of the chloroplast involving both outer and inner envelopes while plastid vesicles seem to originate only from the inner envelope.

The authors agree that movement inside stromules and movement from the inner envelope to the thylakoids cannot directly be compared. Despite this, velocity measurements of transport through plastid tubules provide an interesting idea on how fast molecules are transported within the highly viscous stroma as it was shown by Köhler et al. that diffusion velocities differ between stroma, cytosol and aqueous solutions. This was clarified in the text.

16. Page 14, lines 410-423. The discussion on two putative chloroplast vesicle-associated proteins is confusing. Please rewrite for clarity.

cpSAR1 and cpRabA5e are the only 2 candidate proteins found during bioinformatics approaches with confirmed chloroplast localization so far. In our opinion, it makes no sense to leave one of these proteins out when it comes to giving the state of the art.

17. Page 16, line 485-488. The authors mention “circular membranous structures” but there is no discussion of the fate of these structures during the rehydration of the desiccoplast.

The fate of these structures is not known. Circular membranous structures were observed to be present in most chloroplasts and they are thought to be a consequence of invagination of the boundary membrane. This was clarified in the text.

18. Page 16, line 490. “membrane-bound vesicles”. Are all vesicles “membrane-bound”?

In this particular case no free vesicles were observed that contributed to the formation of pro-thylakoids. “Membrane-bound” was rephrased to “membrane-attached” to avoid confusion.

19. Page 16, line 492. “thylakoid precursor membranes are being formed”. Please clarify from where?

Thylakoid precursor membranes are formed after many of the membrane-bound vesicles start to elongate into prothylakoids in a first step of the xeroplast-chloroplast transition. This was clarified in the text.

20. The authors write on page 16 that the revived thylakoids may form with little contribution from the inner envelope”. Does that mean the desiccoplast development and differentiation is really different from normal chloroplast biogenesis. I agree that the desiccoplasts are a good experimental system on their own right but I am not sure how much we can learn about normal thylakoid biogenesis from this plastids.

As Tuba et al. observe a clear correlation between the sequential disappearance of the membrane-bound vesicles and the appearance of the prothylakoid membranes, they suggest that in contrast to proplastids, the thylakoid membranes in desiccoplasts are not derived from the inner chloroplast membrane. They further suggest that the stockpile of MGDG and DGDG (that are synthesized on the inner and outer chloroplast membranes of proplastids) in the membrane-bound vesicles in desiccoplasts is sufficient for thylakoid assembly. Besides this difference, there is strong evidence that the molecular mechanisms underlying photomorphogenic development in etioplast-chloroplast transition as well as in desiccation tolerance are similar. This was clarified in the text.

21. The opening sentence in “Conclusion” section is confusing. Please rephrase.

The sentence was rephrased.